# Effect of Proton Therapy on Tumor Cell Killing and Immune Microenvironment for Hepatocellular Carcinoma

**DOI:** 10.3390/cells12020332

**Published:** 2023-01-15

**Authors:** Miao-Fen Chen, Ping-Tsung Chen, Ching-Chuan Hsieh, Chih-Chi Wang

**Affiliations:** 1Department of Radiation Oncology, Chang Gung Memorial Hospital at Linko, Taoyuan 333, Taiwan; 2College of Medicine, Chang Gung University, Taoyuan333, Taiwan; 3Department of Medical Oncology, Kaohsiung Chang Gung Memorial Hospital, Kaohsiung 833, Taiwan; 4Department of Surgery, Chang Gung Memorial Hospital at Chiayi, Chiayi 613, Taiwan; 5Department of Surgery, Kaohsiung Chang Gung Memorial Hospital, Kaohsiung 833, Taiwan

**Keywords:** HCC, proton therapy, PD-L1, immune

## Abstract

Radiotherapy with proton therapy (PT) has dosimetric advantages over photon therapy, which helps to enlarge the therapeutic window of radiotherapy for hepatocellular carcinoma (HCC). We evaluated the response of HCC to PT and examined the underlying mechanisms. The human liver cancer cell lines HepG2 and HuH7 and the murine liver cancer cell line Hepa1–6 were selected for cell and animal experiments to examine the response induced by PT irradiation. Biological changes and the immunological response following PT irradiation were examined. In vitro experiments showed no significant difference in cell survival following PT compared with photon radiotherapy. In a murine tumor model, the tumors were obviously smaller in size 12 days after PT irradiation. The underlying changes included increased DNA damage, upregulated IL-6 levels, and a regulated immune tumor microenvironment. Protein analysis in vitro and in vivo showed that PT increased the level of programmed cell death ligand 1 (PD-L1) expressed in tumor cells and recruited myeloid-derived suppressor cells (MDSCs). The increase in PD-L1 was positively correlated with the irradiation dose. In Hepa1–6 syngeneic mouse models, the combination of PT with anti-PD-L1 increased tumor growth delay compared with PT alone, which was associated with increased tumor-infiltrating T cells and attenuated MDSC recruitment in the microenvironment. Furthermore, when PT was applied to the primary HCC tumor, anti-PD-L1 antibody-treated mice showed smaller synchronous unirradiated tumors. In conclusion, the response of HCC to PT was determined by tumor cell killing and the immunological response in the tumor microenvironment. The combination with the anti-PD-L1 antibody to enhance antitumor immunity was responsible for the therapeutic synergism for HCC treated with PT. Based on our results, we suggest that PT combined with anti-PD-L1 may be a promising therapeutic policy for HCC.

## 1. Introduction

Hepatocellular carcinoma (HCC) is a globally occurring cancer with a relatively high incidence in southeast Asia [1]. The majority of patients have a poor prognosis due to advanced tumors and/or poor hepatic function. Traditionally, radiation therapy (RT) with definite dose for the treatment of liver cancer has been associated with a high risk of radiation-induced hepatitis. Improvements in radiotherapy delivery techniques have helped to enlarge the RT therapeutic window for HCC. The development of RT with proton therapy (PT) has brought the use of RT for HCC into the spotlight [2,3]. PT has dosimetric advantages over conventional photon therapy for cancer treatment, especially for HCC, head and neck cancer, and breast cancer. Recent studies suggested that there may be differences in the RT-induced biological changes, including in the induction of DNA damage, oxidative stress, and the regulation of immune cells, between PT and photons at equivalent relative biological effectiveness doses [4,5,6,7]. Further investigation of the radiobiological effect of PT will facilitate the opportunity to provide the development of promising PT strategies for cancer treatment.

The combination of RT and immunotherapy in clinical settings is promising for many indications [8,9]. As chronic inflammation results in a predisposition to the development of HCC, HCC is a potential target for immunotherapy [10]. RT has both proimmunogenic and immunosuppressive effects on the tumor microenvironment [9]. Although RT has been shown to trigger immunogenic cell death and enhance antitumor immune cell infiltration [11,12], several studies showed that RT can induce cytokines and immune regulatory cells, resulting in a more immunosuppressive tumor microenvironment. The combination of immunotherapy and RT was shown to enhance the tumor response compared with either modality alone and to abrogate the RT-induced immunosuppressive effects [13,14]. Several preclinical and clinical studies showed that treatment with photon RT and immunotherapy has a positive effect on tumor control in HCC [15,16]. To our knowledge, almost all reports regarding radio-induced immune responses have been obtained with irradiation based on photons. A better understanding of the mechanisms responsible for the response to PT could lead to more potent therapeutic approaches for HCC. The biological effects of PT responsible for tumor control and the immune response need further investigation. Therefore, we aimed to evaluate the radiation effect with PT and the immune response in the tumor microenvironment. We also investigated whether the combination of PT and the blockade of PD-L1 has a synergistic effect on HCC tumor control.

## 2. Materials and Methods

### 2.1. Cell Culture

The human liver cancer cell line HepG2 and murine liver cancer cell line Hepa1–6 were obtained from the Bioresource Collection and Research Center. In addition, HuH7 cells, a human hepatoma cell line, were kindly provided by Dr. Yen-Hao, Chen (Kaohsiung Chang Gung Memorial Hospital, Taiwan). The use of the cell lines was approved by our institutional research ethics committee (No. 00464-2021120952922). The cell lines were cultured in Dulbecco’s modified Eagle’s medium (DMEM) supplemented with 10% fetal bovine serum (FBS). In addition, human monocytic THP-1 cells were maintained in RPMI 1640 culture medium. THP-1 monocytes can be differentiated into M1 or M2 macrophages when incubated in conditioned medium [17,18]. To determine the effects of PT on the ability of cancer cells to induce M2 macrophage differentiation, HuH7 cells with or without PT irradiation were seeded in 6-well plates 24 h before the end of M2 macrophage polarization. After 24 h of coculture, the expression of CD163, an M2 macrophage marker, was analyzed with macrophages. Multicolor fluorescence-activated cell analysis (FACS) was performed using a FACS caliber flow cytometer (BD Biosciences) on single-cell suspensions prepared from the differentiated macrophages, and immunostaining for CD163 was carried out using a fluorescence-labeled monoclonal antibody. The experiments in vitro were performed in the Conjoint Laboratory of Chang Gung Memorial Hospital, which provided the service platforms including the cell sorter, multicolor cell analyzer, and confocal microscopy. 

### 2.2. Irradiation

The relative biological effectiveness of protons was set at 1.1, and the relative biological effectiveness (RBE) dose was calculated by multiplying the proton dose by this value [19]. In the present study, we used the RBE dose to evaluate the effect induced by RT. Dosage in proton radiotherapy is prescribed as Gy (RBE) by scaling up the physical dose by 10%, i.e., the proton RBE dose = the physical proton dose × 1.1. Cells and mice received local RT either by photon or PT irradiation with the RBE dose. Photon irradiation was performed using Varian 21EX and Eclipse treatment planning. In vitro, exponentially growing cells were irradiated with single doses of 0, 3, 6, and 9 Gy using a 6 MV beam for photon radiation analysis. For PT, irradiation by pencil beam scanning (PBS) was performed at different proton doses (0, 3, 6, 9, and 12 Gy (RBE)), with an energy corresponding to 93.6–109.2 MeV and 72–143.2 MeV for cells and mice, respectively. Cell and mouse irradiations were conducted by placing the cells at the middle of the spread-out Bragg peak (SOBP; 1 cm width for cell PT, 6 cm width for mouse PT irradiation) to simulate clinical conditions (Figure 1a,b). Proton irradiation was delivered by the cyclotron used at our hospital (Sumitomo Heavy Industries, Ltd., Tokyo, Japan), which generates a continuous and high-intensity proton beam. The RT field size was 20 × 10 cm^2^ to examine abscopal effect, 20 × 15 cm^2^ to examine the tumoricidal effect of PT in vivo, or 20 × 20 cm^2^ to examine the effect of PT in vitro. Ray Station was the treatment planning system used for RT dose calculation (version 8.1, Ray Search Laboratories, Stockholm, Sweden).

### 2.3. T-Cell Proliferation Assay

To examine whether PT was able to regulate the effect of cancer cells and myeloid-derived suppressor cells (MDSCs) on CD8+ T-cell proliferation, we measured the proliferation of CD8+ T cells after stimulation. The CD8+ T cells isolated using anti-CD8 microbeads were labeled with carboxyfluorescein succinimidyl ester (CFSE) and seeded in 96-well plates with CD11b+ cells in the presence or absence of cancer cells 48 h after PT. The proliferation of CD8+ T cells was stimulated by anti-CD3/CD28 beads (Invitrogen, Waltham, MA, USA). The CFSE fluorescent staining was analyzed using flow cytometry 3 days after stimulation.

### 2.4. Induction of CD14+HLA-DR—From Peripheral Blood Mononuclear Cells (PBMCs)

A novel subset of MDSCs were identified based on the presence of CD14 expression but the absence of human leukocyte antigen (HLA)- DR expression (CD14+HLA-DR−) in the peripheral blood of cancer patients. The abnormal accumulation of CD14+HLA-DR− cells in PBMC reportedly contributes to tumor immune evasion and correlates with cancer prognosis [20,21,22]. To evaluate the role of PT irradiation in the induction of MDSCs, the percentage of CD14+HLA-DR−myeloid cells was evaluated from PBMCs incubated with PT-irradiated cancer cells for 24 h. The proportion of CD14+HLA-DR− myeloid cells and the expression level of PD-L1 were analyzed using FACS (BD Biosciences, San Jose, CA, USA).

### 2.5. Clonogenic Assay

Clonogenic assays were used to examine the effect of RT (PT versus photon RT) on the loss of reproductive cell survival. The cell cultures were irradiated and then incubated at 37 °C for colony formation. After 10 days, colonies were fixed and stained with crystal violet for colony counting. The colonies were scored to determine plating efficiency and the surviving fractions at a given RBE dose. The survival fraction is the number of colonies after RT exposure, with a correction for the plating efficiency. 

### 2.6. Syngeneic (Ectopic and Orthotopic) Tumor Models

All animal studies complied with all relevant ethical regulations for animal research and were approved by the experimental animal committee of our hospital. The animal experiments were performed in the Laboratory Animal Center of Chang Gung Memorial Hospital, which is granted a full accreditation from the Association for Assessment and Accreditation of Laboratory Animal Care International (AAALAC). We used C57BL/6J mice as the liver tumor implantation model. In the ectopic and orthotopic tumor model, tumor cells (Hepa 1–6 1 × 10^6^ cells) were subcutaneously implanted into the right thigh region and/or intraoperatively implanted into the liver dome area. To examine the response of liver tumor to PT in vivo, local PT irradiation for 12 Gy (RBE) was given to tumors 14 days after implantation with Hepa1–6 cancer cells (Figure 1b). The control mice were subjected to sham irradiation. To address the abscopal effect of PT on tumor-bearing immunocompetent hosts, we simultaneously implanted Hepa1–6 cancer cells into the right thigh (primary tumor) and upper back (secondary synchronous tumor). Fourteen days after tumor implantation, PT with 12 Gy (RBE) was administered to the primary tumor but not to the secondary synchronous tumor. We then observed the tumor growth (including primary irradiated and synchronous unirradiated tumors) at the indicated time points. To investigate the effects of anti-PD-L1 on PT-mediated local tumor control and the abscopal effect, tumor-bearing mice were given an intraperitoneal dose of 250 μg anti-PD-L1 antibody immediately after PT irradiation and every 2 days until the end of the experiments. The anti-mouse PD-L1 (B7-H1, 10F.9G2) antibody was obtained from Bio X Cell (Lebanon, NH, USA).

### 2.7. MDSC Flow Cytometric Analyses In Vivo

MDSC are characterized by co-expression of the myeloid-cell lineage differentiation antigens Gr1 and CD11b. Therefore, we used a specific anti-Gr1 antibody, which reacts with a common epitope on Ly-6G and Ly-6C, and an antibody specific for CD11b (BD Pharmingen) to define mouse MDSCs as CD11b + Gr1+ [23]. Furthermore, myeloid differentiation antigen Gr-1 consists of two epitopes recognized by anti-Ly-6G and anti-Ly6C antibodies. The population of CD11b+Gr-1+ MDSCs consisted of two major subsets: cells with a granulocytic phenotype expressing Ly-6G, and cells with a monocytic phenotype expressing Ly6C. A novel subset of MDSCs was identified as monocytic MDSCs, defined as CD11b+Ly6G- in mice. We performed FACS and immunofluorescence analyses to examine the effect of irradiation on MDSC recruitment after mice received irradiation. FACS was carried out on single-cell suspensions prepared from whole tumors and spleen after digestion and immunostaining for CD11b, Gr1, and LY6G with fluorescence-labeled monoclonal antibodies (BD PharMingen). The percentage of MDSC was measured via multicolor flow cytometry with the abovementioned monoclonal antibodies. Isotype-specific antibodies were used as negative controls in FACS.

### 2.8. Statistical Analysis

Samples were analyzed using Student’s t-test. Data are presented as the mean ± standard error of the mean (SD). All the cellular experiments comprised three biological replicates per condition [24] and were performed at least three times independently. For the in vivo experiments, six animals were used per group, and at least two independent experiments were performed. A probability level of *p* < 0.05 was taken to indicate statistical significance, unless otherwise stated.

## 3. Results:

### 3.1. Response of HCC to PT Response of HCC 

Human and murine cancer cells were exposed to a single PT dose of 0, 3, 6, or 9 Gy (RBE), and the cell death at 48 h and the survival fraction of colony-forming cells were examined and compared with photon-RT. There was no significant difference in cell survival at equivalent RBE between PT and photon-RT (Figure 2a,b). The formation of p-H2AX is a cellular response to DNA double-strand breaks, and the presence of calreticulin exposure and high mobility group Box 1 (HMGB1) serve as the hallmarks of RT-induced immunogenicity cell death [25,26,27]. As shown in Figure 2c, PT enhanced calreticulin and HMGB1 expression associated with increased DNA damage 24 h after RT. A previous study [28] reported that photon-RT upregulated IL-6, which was related to the resistance of HCC to RT. The data (Figure 2c,d) reveal that PT upregulated the level of IL-6 in liver cancer cells analyzed using IF and real-time RT PCR at 24 and 48 h after PT. To further characterize whether the increase in IL-6 by PT was linked with IL6 promoter activity, we performed luciferase activity assays using HepG2 and HuH7 cells stably transfected with a vector expressing IL-6 promoter constructs (Figure 2e). The quantitative data suggest that PT augmented the IL-6 promoter activity at 48 h after 6 Gy (RBE) compared with control cells.

### 3.2. Response to PT Treatment in the Immunocompetent Host

In vivo animal tumor models are essential tools in predicting the efficacy of novel anticancer strategies. We used syngeneic mouse tumor models to investigate the effects of PT on HCC tumor control. Based on the observation of tumor activity via fluorescence molecular tomography (FMT) assay in situ and tumor size measurement, we found that there was significantly decreased tumor glucose-uptake activity and smaller tumors 12 days after PT compared with sham-RT (Figure 3a,b). To further examine the mechanism responsible for the response to PT, we performed FACS and immunofluorescence analyses using tumors 3 days after PT or sham irradiation. It has been reported that the exposure of calreticulin is an immunogenic indicator of lower-dose RT-induced apoptosis, and HMGB1 linked to higher-dose RT-induced cell death [29]. Furthermore, IL-6 has been reported to play a role in RT-induced immune modulation [30]. We previously reported that IL-6 expression was upregulated by photon RT and that the increase of IL6 correlated with the radiation response of liver tumors [28]. As shown in Figure 3c,d, PT increased tumor cell death associated with increased DNA damage, and IL6 and HMGB1 expression compared with sham irradiation.

### 3.3. The Immunomodulatory Effects Induced by PT

The induction of macrophage M2 polarization has been reported to be the key immunosuppressive component in the irradiated tumor microenvironment [17,18]. To test if PT enhanced monocyte differentiation into M2 cells, we incubated THP-1 monocytes at resting stage (M0) in conditioned medium for 72 h for M2 polarization. We then analyzed the levels of the M2 marker in macrophages from monocytes by coculture in culture supernatant with or without PT-irradiated HuH7 cells 24 h before the end of M2 macrophage polarization. As shown in Figure 4a, the addition of PT-irradiated cancer cells to the macrophage culture increased the expression of the M2 marker CD163 in macrophages in vitro. We further researched the role of PT in the ability of cancer cells to induce MDSCs from monocytes in peripheral blood of donors by coculture with cancer cells with or without PT for 48 h. A novel subset of MDSCs were identified by CD14+HLA-DR−cells in the PBMC. Figure 4b reveals that PT irradiation was associated with a higher frequency of CD14+ HLA-DR-myeloid cells in comparison with the control. Furthermore, there was a higher expression of iNOS, a functional marker of MDSC-mediated immunosuppression, in the subset of cells which were cocultured with PT-irradiated cancer cells (Figure 4c). Furthermore, CD163 is confirmed to be a phenotypic marker of M2 macrophages that can be used to distinguish M2 and M1 macrophages. As shown in Figure 4d,e, PT led to the increase of CD163+ cells in PT-irradiated tumors associated with the induction of monocytic MDSCs in tumor-bearing mice. RT has been reported to trigger immunogenic cell death and enhance immune cell infiltration. To further test the role of PT in the functional consequences of cancer cells on MDSC-mediated T-cell suppression, the proliferation of sorted CD8+ T cells was assessed in the presence of tumor cells with or without 6 Gy (RBE) PT irradiation. The data reveals that MDSC-CD11b+ cells decreased T-cell proliferation, and incubation with PT-irradiated tumor cells reversed the proliferation of CD8+ T cells after stimulation (Figure 4f).

### 3.4. Role of PT in the Expression of Programmed Cell Death Ligand 1 (PD-L1)

PD-L1 is a critical determinant of balance in the immune tumor microenvironment [31]. PT increased the expression of PD-L1 in liver tumor cells, and the expression level was positively correlated with the RBE dose of PT in vitro (Figure 5a,b). To further validate the effects of PT on PD-L1 in vivo, we examined the expression level of PD-L1 in murine HCC tumors using IF and FACS. The data in Figure 5c,d reveal significantly increased PD-L1 expression in tumors 3 days following PT. We also examined whether PT regulated the expression of PD-L1 in MDSCs using sorted CD11b+ myeloid cells from tumors. The results indicated that PT increased PD-L1 expression in the sorted CD11b+ cells (Figure 5e,f).

### 3.5. Effect of PD-L1 on the Response of HCC to PT In Vivo

To determine whether PD-L1 plays a role in the radiosensitivity of liver tumors to PT in immunocompetent hosts, local RT with PT 12 Gy (RBE) was administered to subcutaneously implanted tumors in mice with or without anti–PD-L1 treatment. As shown in Figure 6a–c, anti–PD-L1 augmented the PT-induced tumor inhibition effect associated with decreased cell proliferation and augmented cell death after irradiation. Furthermore, to validate the regulatory effects of anti-PD-L1 on the immune tumor microenvironment following PT, we examined the immune response using a murine HCC orthotopic tumor model. The data show that anti-PD-L1 attenuated MDSC recruitment and increased CD3+ TILs associated with smaller tumors compared with PT alone (Figure 6d–f).

### 3.6. Blockade of PD-L1 Enhances the Abscopal Effect on Liver Cancer following PT

Radiation has been reported to induce abscopal effects on distant, untreated cancer sites that are amplified by the use of immunomodulating drugs [32,33,34]. Accordingly, we further examined the abscopal effect induced by PT on liver tumors and the impact of combined treatment with the anti-PD-L1 antibody. As shown in Figure 7a,b and Appendix A, application of local PT to the tumor alone induced smaller tumors with lower glucose metabolism in irradiated tumors over the right thigh, but showed no significant tumor inhibition on the secondary unirradiated tumors in the upper back, compared with the sham-RT group. Combined treatment with the anti-PD-L1 antibody immediately after PT enhanced the tumoricidal effect in the PT-irradiated field and resulted in the regression of secondary tumors outside of the irradiated field. Analysis of tumor-infiltrating immune cells showed that anti-PD-L1 attenuated MDSC recruitment, increased CD3+ TILs, and decreased cell proliferation in unirradiated tumors (Figure 7c,d) of mice that received local PT to primary tumors. These results suggest that anti-PD-L1 enhanced the antitumor immune response and augmented the abscopal effect in immunocompetent hosts following local PT irradiation.

## 4. Discussion

The results of this study show that the loss of colony-forming cells induced by PT is dose-dependent and similar to that caused by photon irradiation. The level of unrepaired DNA damage caused by radiation is a major determiner of the tissue-specific radiation response. Our data reveal that the cell death induced by PT correlated with the increase in the levels of DNA damage markers. Several tumor-associated cytokines are reported to be regulated by irradiation and are able to recruit and polarize immune subsets in the tumor microenvironment [35,36]. We previously reported that IL-6 expression was upregulated by photon RT and that the increase of IL6 correlated with the radiation response of liver tumors [28]. Preclinical studies have shown that there is differential regulation of inflammatory factors after PT versus photon radiation, including for IL-6 [6,7,37]. We show using cellular experiments that PT irradiation also upregulated the expression level of IL-6 in a dose-dependent manner.

Tumor responsiveness to treatment is influenced by tumor cell proliferation and the tumor microenvironment. Using mouse models is an optimal strategy to evaluate treatment response. To our knowledge, few preclinical series have presented the response of liver tumors to PT until now. The in vivo data of tumor cell death and tumor growth delay induced by PT in liver-tumor-bearing mice was demonstrated in the present study. In addition to intrinsic cellular radiosensitivity, tumor regrowth after radiotherapy in vivo may be substantially affected by several inflammatory and stromal factors [38,39]. A variety of immune-related downstream effects are induced by RT, including immunostimulatory and immunosuppressive effects. RT stimulates anticancer immunity through induction of immunogenic cell death, releasing new antigens to the components of the immune system, subsequently leading to improved priming and activation of effector T cells. On the other hand, RT leads to immunosuppressive effects, such as the recruitment of MDSCs to the irradiated microenvironment [39,40]. The recruitment of MDSCs is a determinant of the immunosuppressive tumor microenvironment following RT. MDSCs have emerged as major regulators of immune responses in cancer and other pathological conditions. Evidence supports key contributions of MDSC to tumor progression. We previously reported that increased MDSC could be triggered by RT and that the increase was associated with RT resistance in an HCC animal model [28]. The presented study demonstrated that PT leads to the activation of MDSC recruitment associated with the induction of monocytic MDSCs in tumor-bearing mice. A novel subset of MDSCs identified as monocytic MDSCs are defined as CD14+HLA-DR− monocytes from the peripheral blood of patients and CD11b+Ly6G- in mice [20,23,41]. The expression of iNOS as a functional marker of T-cell inhibition is the gold standard for evaluation of MDSC function [42]. Experiments using PBMC incubation with cancer cells showed that PT enhanced the ability of cancer cells to induce MDSCs in monocytes and increased the expression of iNOS in the subset of cells. Furthermore, our data reveal that coculture with CD11b+ cells decreased T-cell proliferation, and PT-irradiated cancer attenuated the suppressive ability of CD11b+ cells on T-cell proliferation in coculture experiments.

The myeloid cell lineage is reported to constitute a network of immune suppressive cells that are present in most cancer patients and which profoundly inhibit the generation of antitumor immunity. Macrophages, the abundant immune cells in the tumor microenvironment, are important regulators of chronic inflammation. It has been suggested that macrophages can be polarized toward the M2 phenotype, which contributes to the immunosuppressive microenvironment [43]. Mononuclear cells in tumors likely exist in various differentiation phases from monocytes/monocytic-MDSCs to tumor-associated M2 macrophages. This network between MDSCs and tumor-associated M2 macrophages reveals that M2 can be distinguished from monocytic MDSCs. Evidence suggests that RT plays a regulatory role in macrophages with M1/M2 polarization to modulate the immune response [44,45]. We showed that coculture with PT-irradiated cells resulted in increased expression of M2 markers in vitro and PT-irradiated tumors in vivo. The activation of anticancer immune responses is critical to the effectiveness of RT. 

Multiple immune mechanisms are important in the development and progression of HCC and correlate with prognosis. Checkpoint inhibitors targeting PD1 and PDL1 are active, tolerable and clinically beneficial against advanced HCC [46]. PD-L1, a major cellular biomarker, plays a role in immune evasion, including the inhibition of T-cell-mediated immune surveillance and regulation of macrophage M2 polarization [13,31,47]. Upregulation of the PD-1/PD-L1 axis was observed to suppress the cytotoxic action of T cells, which may be the cause of tumor-evading host immune responses and incomplete tumor cell killing after irradiation. PD-L1 was widely located in hematopoietic cells, including T cells, macrophages, and tumor cells. PD-1/PD-L1 is a notable immune checkpoint leading to T-cell anergy. Antibodies to PD-L1, an immune checkpoint blockade, have been approved for adjuvant therapy of some cancers and as a promising immunotherapy for advanced HCC patients [48,49]. PDL1 inhibitors are one of the backbones of systemic therapies in clinical practice or under development for HCC. Furthermore, the RT-induced immunosuppressive effects reveal the potential for successfully combining radiation with various forms of immunotherapy to abrogate these immunosuppressive effects. MDSCs can contribute to patient resistance to immune checkpoint inhibition. We demonstrated that PT upregulated the expression of PD-L1 in tumors and MDSCs in a dose-dependent manner in vitro and in vivo. The RT-induced immunosuppressive effects reveal the potential for successfully combining radiation with various forms of immunotherapy to abrogate these immunosuppressive effects. Preclinical research has shown that the combination of photon RT and immunotherapy exhibits therapeutic synergism to improve tumor control [15]. As we know, the combination of RT and immunotherapy is still in its preliminary stages, and there are no optimal specifications of RT dose, RT types, or sequencing for RT and immunotherapy. To clarify this issue of PT irradiation combined with immunotherapy, we combined anti-PD-L1 therapy with PT irradiation of liver tumors in syngeneic tumor models. We found that the combination with anti-PD-L1 antibody sensitized liver cancer to PT irradiation, as demonstrated by a smaller tumor associated with augmented tumor cell death. Furthermore, anti-PD-L1 therapy inhibited MDSC recruitment and increased the filtration of CD3+ T cells in PT-irradiated tumors. 

RT is classically thought of as a local therapy, killing tumor cells via intrinsic DNA damage in the radiation field. Additionally, the abscopal response to radiation is a well-established phenomenon where tumors outside a radiation field also respond to treatment. The abscopal effect is thought to be immune mediated. A portion of the cancer cells within a tumor will die via immunogenic cell death when radiation is used at therapeutic doses. The RT-induced tumor cell death is associated with the generation of specific molecular signals and more antigens being presented to the components of the immune system. It has been reported that RT may trigger an immune-related abscopal effect, which implies that RT not only has a tumoricidal effect on the target tumor but also has an antitumor effect on distant, untreated sites of cancer [32,50,51]. Among the RT-induced factors, an ability of radiation to promote recognition of cancer cells by T cells is likely of particular importance for the abscopal effect. This effect is associated with tumor cell death to stimulate antitumor immunity that can be amplified by the use of immunomodulating agents. Furthermore, pembrolizumab produces a 15–20% rate of objective remissions that are associated with prolonged survival in HCC [46]. One critical point regarding checkpoint inhibitors is their efficacy and safety. It is clear that immunotherapy does not provide benefit in all patients. How to overcome the tumor resistance to immunotherapy is an important issue. The potential causes of tumor resistance include intrinsic resistance and the development of anti-drug antibodies that neutralize the activity of immunotherapy. To obtain a strong immune stimulation, different local therapies can be combined sequentially or simultaneously with systemic immunotherapy. Immunotherapy is likely to synergize with local interventions in HCC. HCC is often multifocal with potential precancerous areas developing as metachronous throughout. Here, we further examined whether the combination of PT with anti-PD-L1 enhances the abscopal effect of PT to increase the control of metachronous liver tumors. Our data reveal that the combination of PT with anti-PD-L1 induced smaller tumor sizes in the secondary unirradiated tumors compared with those induced by either PT or anti-PD-L1 alone. Our data also show that the combination with anti-PD-L1 was associated with augmented TILs and attenuated MDSC recruitment in distant unirradiated tumors of mice that received local PT. These results suggest that anti-PD-L1 augmented antitumor immunity, which mediated the increased primary tumor-killing activities and the abscopal effect in tumor-bearing mice that received local PT irradiation.

## 5. Conclusions

The rationale for combining radiotherapy and immunotherapy is that radiation could produce synergistic antitumor immunity and that immunotherapy overcomes the immune suppression responses in the irradiated tumor microenvironment. In summary, we suggest that in addition to the biological effect on cancer cells, PT also has an immune modulation effect on the tumor microenvironment. Our data reveal that anti-PD-L1 elicited anticancer immunity and subsequently augmented the response of primary and distant tumors to PT irradiation. The application of anti-PD-L1 combined with PT could be a promising strategy for the treatment of HCC.

## Figures and Tables

**Figure 1 cells-12-00332-f001:**
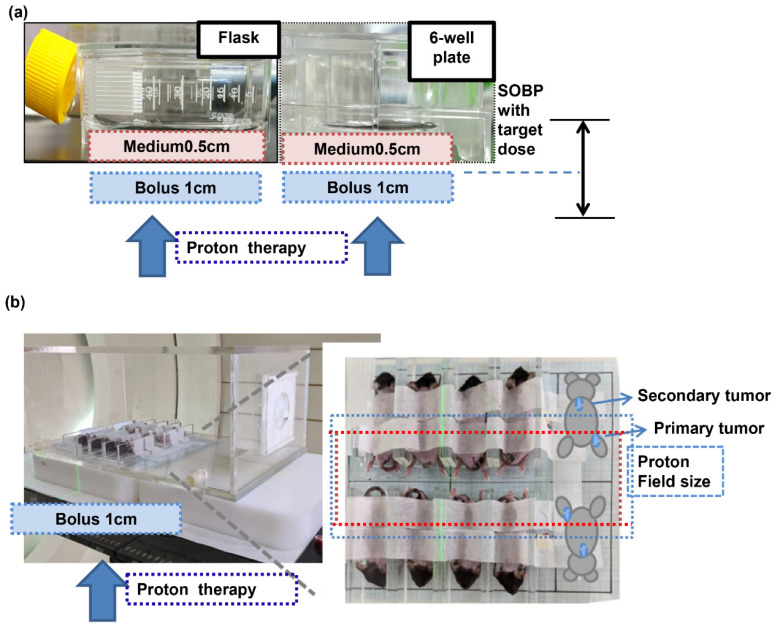
Preclinical model for PT. The experimental proton beam designs for in vitro (**a**) and animal tumor models (RT field for liver tumor irradiation, blue line; RT field to examine abscopal effect, red line) (**b**).

**Figure 2 cells-12-00332-f002:**
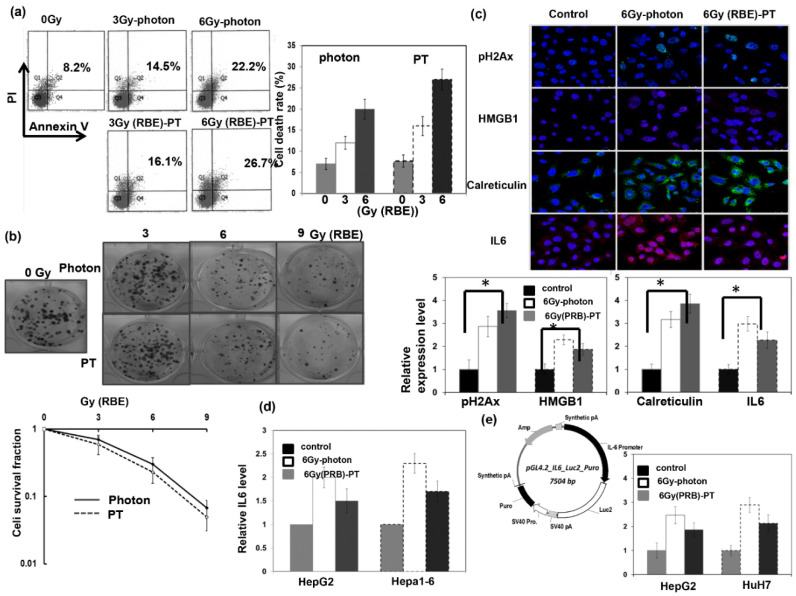
Response of HCC to PT in vitro. Effects of PT treatment on the cell death of HuH7 cells. (**a**) Fluorescence-activated cell sorting with propidium iodide and Annexin V staining 48 h after PT for the indicated dose (RBE), and (**b**) cell survival fractions by clonogenic assays presented with the ratio normalized by the survival fraction under control conditions. The plating cell numbers for 0, 3, 6, and 9 Gy (RBE) were 500, 1000, 1500, and 3000 per well, respectively. (**c**) Immunofluorescence for PT-induced DNA damage and markers for immunogenicity cell death are shown by representative slides and quantitative data (DAPI, blue; pH 2AX and calreticulin, green; HMGB1 and IL6, red). The *y* axis represents the relative fold changes in the target protein expressions. (**d**) The levels of IL-6 were examined via real-time RT–PCR 48 h after PT. (**e**) Stable transfection of HepG2 and HuH7 cells with plasmids containing IL-6 promoter–reporter constructs. Values are represented as fold activation of luciferase activity of the reporter plasmid in the indicated condition. Data are presented as means ± standard errors of the mean. * *p* < 0.05.

**Figure 3 cells-12-00332-f003:**
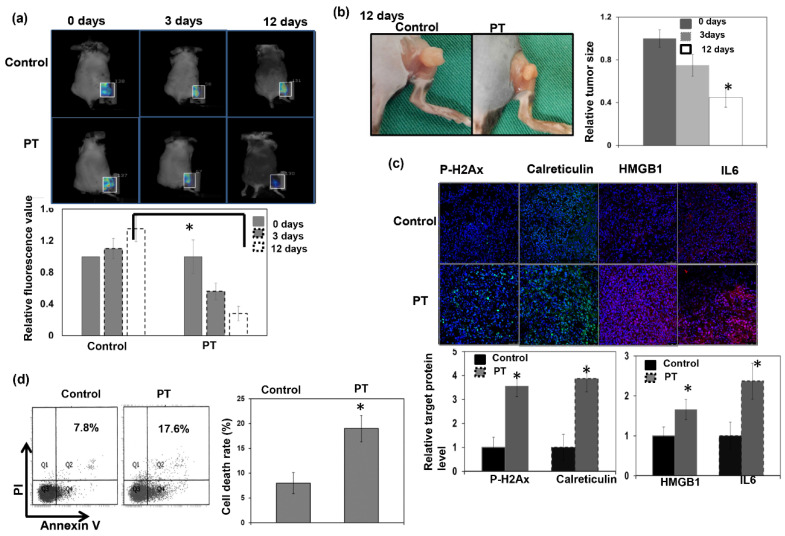
Response to PT treatment in immunocompetent mice. Representative images and quantitative data were determined via FMT analysis of glucose uptake at 0, 3, or 12 days with or without PT irradiation (**a**). The *y* axis represents the ratio normalized by the value of tumor at 0 day with sham irradiation. Representative images and quantitative data of tumor pictures (**b**) from mice bearing s.c. tumors 12 days after 12 Gy (RBE) PT or sham irradiation are shown. The *y* axis represents the relative fold change in tumor size at the indicated time after PT. DNA damage by IF (**c**) and cell death by FACS (**d**) were evaluated at 3 days after PT. The *y* axis is the relative fold change in target protein expressions and the cell death rate at 3 days after PT. Data are presented as means ± standard errors of the mean. * *p* < 0.05.

**Figure 4 cells-12-00332-f004:**
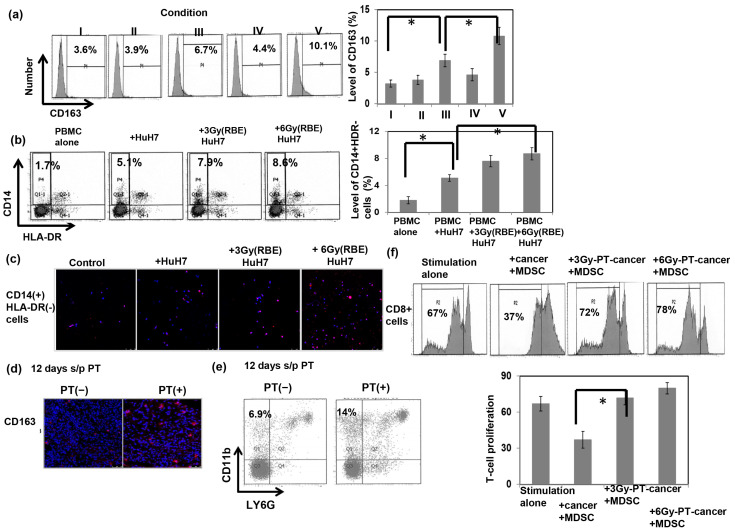
Immune response associated with PT. The expression of the CD163 marker in the cells at the end of M2 macrophage polarization was analyzed via FACS (**a**) (I: control; II: + cancer cell culture supernatant; III: + cancer cells; IV: +6 Gy (RBE) PT-irradiated cancer cell culture supernatant; V: +6 Gy (RBE) PT-irradiated cancer cells). The percentage of CD14+HLA-DR− cells from PBMCs incubated with or without HuH7 cells irradiated with 0, 3, or 6 Gy (RBE) for 48 h was analyzed (**b**), and the expression of iNOS in the sorted CD14+HLA-DR− cells was analyzed via IF (DAPI, blue; iNOS, red) (**c**). The extent of CD163+ cells in irradiated tumors 12 days after PT was examined via IF (DAPI, blue; CD163, red) (**d**), and the effect of PT irradiation on monocytic-MDSC recruitment was evaluated via FACS (**e**). Furthermore, the rate of T-cell proliferation was examined via FACS with or without incubation with PT-irradiated cancer cells. Representative images and quantitative data are shown (**f**). Treatment indicated CD8+ T cells with anti-CD3/CD28 stimulation beads in the presence of MDSC combined with cancer cells with or without PT irradiation. Data are presented as means ± standard errors of the mean. * *p* < 0.05.

**Figure 5 cells-12-00332-f005:**
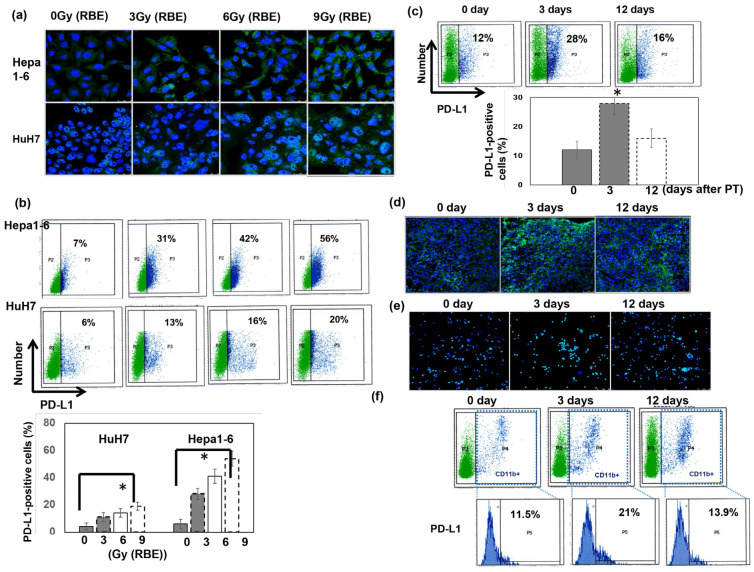
Effect of PT on PD-L1 expression. The levels of PD-L1 were evaluated using (**a**) IF (DAPI, blue; PD-L1, green), and (**b**) FACS staining for human and murine liver cancer cells at 48 h after PT in vitro and (**c**) FACS and (**d**) IF for murine liver tumors at the indicated times after PT for 12 Gy (RBE) in vivo. The *y* axis is the relative fold change in PD-L1 expression at the indicated conditions after PT. Data are presented as means ± standard errors of the mean. * *p* < 0.05. Furthermore, the expression of PD-L1 in murine CD11b+ cells following PT irradiation was evaluated via (**e**) IF and (**f**) FACS analysis (CD11b+ cells, blue spots).

**Figure 6 cells-12-00332-f006:**
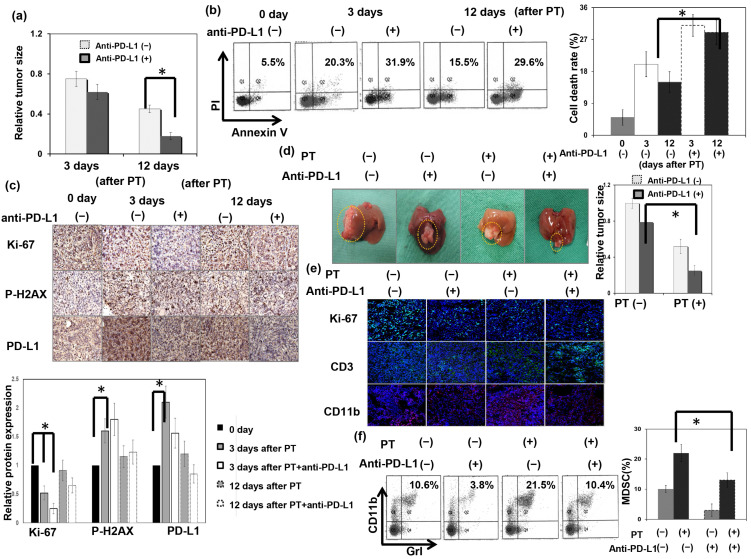
Effect of PD-L1 on the response of HCC to PT in vivo. (**a**) The effect of anti-PD-L1 therapy on the tumor growth delay. The *y* axis is the relative fold change in subcutaneous tumor size induced by anti-PD-L1 after PT. Data represent the means of experiments, * *p* < 0.05. (**b**) The in vivo effects of treatment-induced apoptosis as evaluated using FACS with Annexin V-PI staining. The *y* axis is the relative fold change in the cell death rate after PT. Data are presented as means ± standard errors of the mean. * *p* < 0.05. (**c**) Immunohistochemistry for RT-induced DNA damage and Ki-67 in tumors at the indicated times after PT for 12 Gy (RBE). (**d**) The effect of anti-PD-L1 therapy on tumor inhibition was evaluated in an orthotopic tumor model. Representative images and quantitative data are shown 12 days after local PT irradiation or sham irradiation. The *y* axis is the relative fold change in liver tumor size. Data represent the means of experiments, * *p* < 0.05. (**e**) The extent of ki67+, CD3+TIL, and CD11b+ cells in irradiated tumors 12 days after PT were examined using IF (DAPI, blue; ki-67 and CD3, green; CD11b, red). (**f**) The effect of anti-PD-L1 combined with PT irradiation on MDSC recruitment was evaluated using FACS.

**Figure 7 cells-12-00332-f007:**
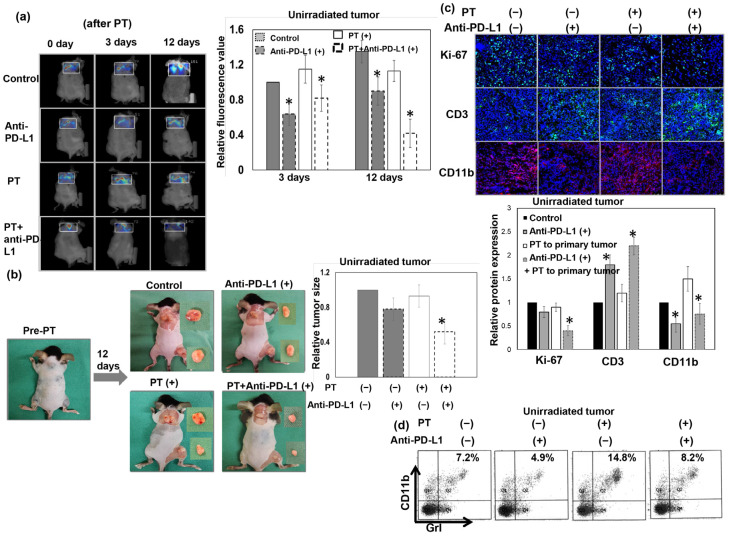
The abscopal effect on liver cancer following PT. The growth of unirradiated tumors over the upper back was determined via FMT analysis of glucose uptake (**a**) and tumor pictures (**b**) at indicated times from mice that received PT for 12 Gy (RBE) to the primary tumor over the right thigh only (PT (+): the mice received 12 Gy (RBE) to the primary tumor over right thigh, but not to the secondary tumor over the upper back). Representative images and quantitative data of unirradiated tumors are shown at 0, 3, and 12 days after PT irradiation with or without anti-PD-L1 therapy. The *y* axis represents the relative fold change in the value of FMT and tumor size at the indicated time after PT. Furthermore, representative images of the level of ki-67 and the infiltration of CD11b+ cells and CD3+ cells in secondary unirradiated tumors 12 days after local PT using IF are shown (DAPI, blue; ki-67 and CD3, green; CD11b, red) (**c**). The accumulation of MDSCs (**d**) in the secondary unirradiated tumor was analyzed using flow cytometry. The *y* axis is the relative fold change. Data represent the means of experiments, * *p* < 0.05.

## Data Availability

Not applicable.

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
