# Peer review of "Effect of Proton Therapy on Tumor Cell Killing and Immune Microenvironment for Hepatocellular Carcinoma"

_cells, 2023, doi:10.3390/cells12020332_

Round 1

Reviewer 1 Report (Previous Reviewer 1)

In this manuscript, authors attempt to present radiation response of hepatocellular carcinoma (HCC) to proton therapy (PT) as a therapeutic strategy for the treatment of HCC. Authors tried well presenting a revised manuscript with necessary changes as suggested.

Overall, the article is nicely written with sound performed descriptive experiments with scientific soundness.

Before proceeding further, I expect the authors to thoroughly proofread the document once again and fix all grammatical and typographical errors.

Minor suggestion:  L177, either provide complete statement or remove it if not necessary. 

Author Response

Reviewer 2 Report (New Reviewer)

The manuscript reports unwanted immune suppression responses of hepatocellular carcinoma by treatment with proton therapy, and one possible solution by the application of anti-PD-L1 combined with PT. I think it is an interesting story. My comments are below.

1) At line 39, “RT” can be “radiation therapy (RT)”.

2) At lines 88 and 89, “relative biological effectiveness dose (RBE)” can be “relative biological effectiveness (RBE) dose”.

3) Regarding anti-PD-L1, is it anti-mouse PD-L1? In addition, you might want to provide more information about the antibody that you used, such as vender, structure, etc, because it is one of the key elements for the report.

Author Response

Reviewer 3 Report (New Reviewer)

The article of Chen et al. was aimed to evaluate the radiation response of hepatocellular carcinoma to proton therapy and the underlying mechanisms associated with this therapy. The authors used human liver cancer cell lines HepG2 and HuH7, the murine liver cancer cell line Hepa1-6  for cell experiments, doubled with animal experiments to examine the response induced by proton therapy irradiation, examining the biological changes and the immunological response following proton therapy irradiation. The response of hepatocellular carcinoma to proton therapy was determined by tumor cell killing and the immunological response in the microenvironment. The combination with anti-PD-L1 to enhance antitumor immunity was found responsible for the therapeutic synergism for hepatocellular carcinoma to proton therapy. Based on presented results, it was suggested that proton therapy combined with anti-PD-L1 may be a promising therapeutic policy for hepatocellular carcinoma. The conclusions are supported by the ample data provided and I recommend publication with the minor corrections indicated below:

Figures should be improved to make them more readable, especially the groups of data:

Figure 2 should be improved: the cell death rate/Gy graph can be made a bit smaller; adjust labels of figure 2c to the left, they are pushed over the graph; the label “Figure 2” should be deleted and Figure 2a should be (raised) aligned with 2c; figure 2b should be made a bit bigger after doing these corrections

Figure 3 presentation should be improved too as it is difficult to follow; efforts should be made to match the size of panels and to group them better

Labels “Figure X” on figures should be deleted

General: the results section should be slightly improved by presenting the rationale of performing the experiments in greater detail; now it is almost a telegraphic enumeration of the experiments performed.

Additional efforts should be made in the discussion section to put the work in larger context of anti-PD-L1 therapy, especially its immune-related limitations in cancer, putting the new (promising) data presented in a better context.

Author Response

Reviewer 4 Report (New Reviewer)

After carefully examining manuscript entitled as " Effect of proton therapy on tumor cell killing and immune microenvironment for hepatocellular carcinoma" written by Chen et al., is very well organized and formatted. The manuscript can be further improved after minor revision before publication in the journal.

1.       Page 1, line 16, delete the word “Material and Methods” in the abstract

2.       Page 1, line 42, write full abbreviation of the RT in the text.

3.       Page 3, line 106. Delete the word “Figure 1”. Similarly, page 6, delete the word “Figure 2”

4.       Add one paragraph about general instrumentation with details information

5.       The test used in Figure 1 is not uniform. Would be better to keep one format and one font size.

6.       Conclusion is missing at the end of Discussion, which need to added as a separate heading.

7.       Some of the references are not according to the journal style and need to re-check and format according to the journal style.

8.     The journal names in the reference section are not italic, which needs to be italic

Author Response

This manuscript is a resubmission of an earlier submission. The following is a list of the peer review reports and author responses from that submission.

Round 1

Reviewer 1 Report

In this manuscript, authors attempt to present radiation response of hepatocellular carcinoma (HCC) to proton therapy (PT) as a therapeutic strategy for the treatment of HCC. Although, authors tried well presenting a revised manuscript with necessary changes as suggested, however, they still lack the repetition sets of experiments. Here, experiments were performed only ‘twice’; however, according to the publication guidance number of performed experiments should be more, for their work to be reproducible and considered reliable (refer following reference). This manuscript does not satisfy the format of a scientific paper and is found incomplete again.

Reference:

Curtis, M. J., Bond, R. A., Spina, D., Ahluwalia, A., Alexander, S. P., Giembycz, M. A., ... & McGrath, J. C. (2015). Experimental design and analysis and their reporting: new guidance for publication in BJP. British journal of pharmacology172(14), 3461.

Reviewer 2 Report

The authors have sucessfully addressed my main concerns.